# Body Weight May Have a Role on Neuropathy and Mobility after Moderate to Severe COVID-19: An Exploratory Study

**DOI:** 10.3390/medicina58101401

**Published:** 2022-10-06

**Authors:** Ignacio Figueroa-Padilla, Dalia E. Rivera Fernández, Erick F. Cházaro Rocha, Alma L. Eugenio Gutiérrez, Kathrine Jáuregui-Renaud

**Affiliations:** 1Unidad de Investigación Médica en Otoneurología, Instituto Mexicano del Seguro Social, Ciudad de México 06720, Mexico; 2Hospital General Regional 72, Instituto Mexicano del Seguro Social, Tlalnepantla de Baz 54030, Mexico

**Keywords:** COVID-19, mobility, muscle strength, neuropathy, weight loss

## Abstract

*Background and Objectives*: Among the extra-pulmonary manifestations of COVID-19, neuromuscular signs and symptoms are frequent. We aimed to assess the correlation between neuromuscular abnormalities (electrophysiological) and mobility measures (Berg Balance Scale and Timed-Up-and-Go test) twice, at least 6 weeks after hospital discharge and 6 months later, taking into account cognitive performance, nutrition, muscle strength, and submaximal exercise capacity. *Materials and Methods*: 43 patients (51.4 ± 9.3 years old) accepted to participate in the study; they had a dyspnea score ≤ 3 (Borg scale), and no history of neurology/neuromuscular/orthopedic disorders, but high frequency of overweight/obesity and weight loss during hospital stay. The two evaluations included physical examination, cognitive assessment, nutritional evaluation, muscle strength (hand-grip and quadriceps dynamometry), electromyography, Barthel Index, Six-Minute- Walk-Test (6MWT), Berg Balance Scale and Timed-Up-and-Go test. Bivariate and repeated measures covariance analyses were performed (significance level of 0.05). *Results*: Electrophysiological abnormalities were evident in 67% of the patients, which were associated with diminished performance on the 6MWT, the Berg Balance Scale and the Timed-Up-and-Go test. At each evaluation and between evaluations, scores on the Berg Balance Scale were related to the body mass index (BMI) at hospital admission and the 6MWT (MANCoVA R ≥ 0.62, *p* = 0.0001), while the time to perform the Timed-Up-and-Go test was related to the electrophysiological abnormalities, weight loss during hospital stay, sex, handgrip strength, and the 6MWT (MANCoVA, R ≥ 0.62, *p* < 0.0001). We concluded that, after hospital discharge, patients with moderate to severe COVID-19 may have neuromuscular abnormalities that can be related to BMI/weight loss, and contribute to mobility decrease. In patients with moderate to severe COVID-19 and high BMI/ large weight loss, neuromuscular and intended mobility assessments could be required to provide early rehabilitation. Apart from the 6MWT, handgrip dynamometry and the Timed-Up-and-Go test were useful tools to quickly assess fitness and mobility.

## 1. Introduction

COVID-19 is caused by the beta coronavirus SARS-CoV-2 [1], and is now recognized as a multi-organ disease [2]. In the acute phase, it can present with a variety of symptoms, with potential involvement of almost all organs and systems [3]. In adults, hospitalization with COVID-19 is associated with high rates of morbidity. The disease may progress to a hyperinflammatory and hypercoagulable state, resulting in a wide range of complications that include, among others, severe acute respiratory distress syndrome, acute kidney failure, deep vein thrombosis and pulmonary embolism, stroke, as well as several cardiovascular complications and sequels, such as myocardial injury, heart failure, heart attack, myocarditis, arrhythmias, blood clots and acute coronary syndrome [3,4]. 

After the acute illness, which usually lasts 4 weeks from the onset of symptoms, COVID-19 survivors may also show residual symptoms and organ impairment that can persist beyond 12 weeks of the onset of disease [5]. In order to understand patient needs and to provide multi-disciplinary care and rehabilitation, methodical study of the sequels is required.

A systematic review and meta-analysis of 350 studies on the neurological manifestations related to COVID-19, including 145,721 patients, showed 41 neurology manifestations; among them, the most common symptoms were fatigue (32%) and myalgia (20%) [6]. In the United States of America, fatigue was reported by 71% of 274 patients with mild disease [7]; while in Italy it was reported by 53.1% of 143 hospitalized patients [8]. In China, a meta-analysis of 55 clinical studies recounted myalgia in 21.9% of 8697 patients [9]. A multicenter European study revealed myalgia in 62.5%, and asthenia in 63.3%, of 1420 patients who had no need of intensive care [10].

After the acute care of COVID-19, the high prevalence of neuromuscular symptoms persists. In Australia, after 6 months of critical illness, 21.7% of 112 surviving patients reported loss of strength [11]. In China, at 6 months after hospital discharge, the most frequent symptoms in 1733 patients were muscle weakness or fatigue (63%); also, 25% of the patients had low performance on the 6 Minute Walk Test (6MWT) [12]. The 6MWT is a recognized tool to assess moderate exercise capacity [13], and functional status related to COVID-19 [14]. A systematic review of nine studies assessing functional ability after COVID-19 showed reduction in performing activities of daily living, regardless of the applied scales [15]. In Denmark, among 30-day survivors of COVID-19, 72.1% and 92.6%, returned to work after 1 month and 6 months, respectively, following hospitalization [16]. 

The abilities required to independently care for oneself (activities of daily living) include those activities that enable basic survival and well-being (basic activities) and those required to support daily life at home and in the community (instrumental activities) [17]. However, these activities are supported by mobility (motor activities such as walking, reaching and climbing stairs) [18]. Relevant factors that are related to individual functional ability include cognitive performance [19], nourishment [20], and mobility [18]. In England, a cross-sectional study of 386 patients with COVID-19 showed significant cognitive deficits, after controlling for age, gender, education level, income, racial-ethnic group, pre-existing medical disorders, tiredness, depression and anxiety [21]. In France, among 288 patients who were at home 30 days after hospital discharge, 47.2% had malnutrition [22]. 

Although mobility is a fundamental factor to undertake daily life activities, we did not find studies on the contribution of neuromuscular abnormalities (by electrophysiology) to functioning after acute COVID-19, or the cofactors that could have influence on the relationship between these two variables. Then, we primarily designed a study to assess the relationship between neuromuscular abnormalities (by electrophysiology) and mobility measures (Berg Balance Scale and Timed-Up-and-Go test), at least 6 weeks after hospital discharge due to COVID-19, with a follow-up assessment six months afterwards, taking into account the cognitive performance, nutrition, muscle strength, and submaximal exercise capacity of the patients. The secondary aim of the study was to identify among these cofactors, those that may contribute to the neuromuscular abnormalities and the mobility of patients discharged from hospitalization due to moderate to severe COVID-19.

## 2. Materials and Methods

### 2.1. Participants 

In a general hospital reconfigured to address the surge of patients with COVID-19 during the first two waves of contagions (up to April 2021), 43 consecutive patients (mean age 51.4 ± 9.3 years; 25 men/18 women) who were alive at least 6 weeks after hospital discharge, and fulfilled the selection criteria, accepted to participate in the study. Their general characteristics are described in Table 1. To prevent increased variability due to limited respiratory capacity and neurological or muscular disorders, we verified the main selection criteria at inclusion in the study, namely, that all participants had a maximum dyspnea score ≤ 3 in the Borg scale [23], and none had a history of neurology or neuromuscular or orthopedic disorders (clinical records and interview). 

Between the two study evaluations, three patients declined participation due to change of residence; they were 53 to 69 years old (2 men/1 woman). Therefore, 43 patients were evaluated at the time of inclusion in the study, and 40 patients at follow-up. A sample size of 41 participants was calculated to detect a statistically significant Pearson’s correlation coefficient of 0.5, considering bilateral type I error of 0.01 and type II error of 0.2.

According to the institutional recommendations, during hospitalization, the medical treatment of the participants included dexametasone and antibiotics (ceftriaxone (69%), levofloxacin (37%), alone or combined with azitromicine, claritromicine, menopenem, doxiciclinae or cefotaxam); only four patients had severe disease and required neuromuscular blockade related to mechanical ventilation.

### 2.2. Procedures

A detailed description of the procedures is provided in Appendix A. Evaluations were performed two times, with six months in between (Figure 1). To assess both the main variables and the potential confounders, at each evaluation, after physical examination, the following assessments were performed (within two days): General cognitive performance, by the Mini-Mental State Examination [24];Nutrition, by the Nutritional Risk Screening [25] or the Mini Nutritional Assessment [26] according to age, and by body mass index (BMI), which was estimated as kg/m^2^;The abilities required to independently care for oneself, by the Barthel Index for daily life activities [27];Neuromuscular symptoms, by an in-house short questionnaire to recall symptoms during hospital stay (only at the first evaluation), and to report them if present during the previous week (at the two evaluations), including myalgia, fatigue, muscle spasms/twitches/tremors, and numbness/tingling/burning sensations;Muscle strength, by three measures: (1) the Medical Research Council scale (MRCs) (used with the permission of the Medical Research Council, MRC 1976) [28]; (2) quadriceps isometric strength (Baseline, Back–Leg–Chest dynamometer, White Plains, NY, USA), and (3) handgrip strength (Camry Electronic Hand Dynamometer EH101, South El Monte, CA, USA), including normalized dynamometry measurements per body mass (right and left average (kg)/body mass (kg));Electrophysiological abnormalities of upper and lower limbs, by electromyography records (Nihon Kohen MEB-9400, Japan) by a standardized protocol [29,30]; abnormalities were evaluated by two independent reviewers, according to the guidelines of the American Association of Neuromuscular and Electrodiagnostic Medicine [31];Submaximal exercise capacity by the Six Minute Walk Test (6MWT) [13];Mobility by two measures: (1) the Berg Balance Scale [32] and (2) the modified Timed-Up-and-Go test [33].

After the first evaluation, the patients received rehabilitation according to their individual capacity [34].

### 2.3. Statistical Analysis 

After assessing the data distribution (Kolmogorov–Smirnov test), “t” test (either unpaired or paired) was used to assess comparisons between evaluations and between patients with/without electrophysiology abnormalities; the Pearson’s correlation coefficient was used to explore correlations. Accordingly, repeated measures analyses were performed using Multivariate Analysis of Covariance on the scores of the Berg Balance Scale and the time to perform the Timed-Up-and-Go test. The significance level was set at 0.05. 

## 3. Results

### 3.1. Bivariate Analysis

#### 3.1.1. Cognitive Performance

At the two evaluations, all the participants had an adequate performance on the Mini-Mental State Examination (score ≥ 24); the score range was 25 to 30 at the first evaluation, and 28 to 30 at the second evaluation.

#### 3.1.2. Nutritional Assessment 

At the two evaluations, none of the participants was at risk of malnutrition. However, according to the medical records, at hospital admission, 84% of the patients had BMI ≥ 25; while, at hospital discharge, 100% of the patients had lost weight (mean 10.7%, range 1.7%–26.7%). 

At the first evaluation of the study, 14 of the 43 (32%) patients had BMI ≥ 30 (obesity), and 18 (41%) patients had BMI > 25 ≤ 30 (overweight); at the second evaluation, 17 of 40 (42%) patients had BMI ≥ 30, and 15 (37%) patients had BMI > 25 ≤ 30. Then, the proportional gain at the first evaluation (following hospital discharge) was from 0% to 62% and, at the second evaluation, the gain was from 0 to 70%, with a significant weight increase between evaluations (paired “t” test, t = 3.93, *p* = 0.0003).

#### 3.1.3. Neuromuscular Symptoms (Figure 2):

At hospital admission the most frequent symptoms were fatigue (95%, 95% C.I. 89%–100%), and myalgia (72%, 95% C.I. 59%–85%); while 16% (95% C.I. 5%–27%) of the patients reported muscle spasms/ twitches/tremors, and 25% (95% C.I. 12%–38%) of them reported numbness/ tingling/burning sensations;At the first evaluation, 77% reported fatigue (95% C.I. 63%–89%) and 44% reported myalgia (95% C.I. 29%–59%), while circa one third of patients reported muscle spasms/ twitches/ tremors as well as numbness/ tingling/ burning sensations;At the second evaluation, the frequency of the symptoms decreased. At the two evaluations, men reported symptoms more frequently than women (Table 2).

**Figure 2 medicina-58-01401-f002:**
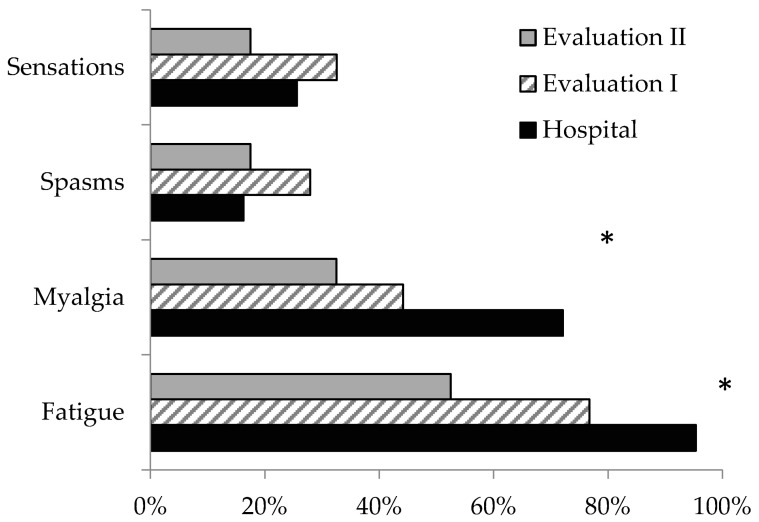
Frequency of symptoms during hospital stay, and two follow-up evaluations of 40 patients with moderate to severe COVID-19. * marked differences are for *p* < 0.05.

#### 3.1.4. Muscle Strength

The results on the modified MRCs are shown in Table A1 of Appendix B, and the dynamometry results are shown in Table 3. At the first evaluation, sarcopenia was diagnosed in six (14%) patients (age 43–65 years, five men/one woman), who recovered at follow-up. Comparison between the two evaluations showed improvement on both the absolute and normalized quadriceps strength (paired “t” test, t values from 3.88 to 4.73, *p* < 0.001), with no change on the handgrip strength (*p* >0.05). 

#### 3.1.5. Electromyography

The main results are shown in Table 4. Normal electromyography recordings were obtained in 15 of the 43 (33%) patients at the first evaluation, and 16 of the 40 (40%) patients at the second evaluation. At the two evaluations, there was no evidence of denervation in any patient. Abnormalities were as follows (Figure 3):Sensory–motor polyneuropathy was diagnosed in seven (16%) patients, including two of the patients who abandoned the study. Three patients had a history of type 2 diabetes, and one was a blacksmith. The recordings were similar at the two evaluations;Motor polyneuropathy was diagnosed in two (4%) patients; one of them with type 2 diabetes, and the other patient required intensive care during their hospital stay, with 45 days of mechanical ventilation. The recordings were similar at the two evaluations;Multiple mononeuropathy (median and peroneal nerves) occurred in four (9%) patients, including one of the patients who abandoned the study. Among the four patients, one had a history of diabetes, and two reported the occupational performance of repetitive hand movements. The recordings were similar at the two evaluations.Mono-neuropathy affecting:○Median nerves at the level of the carpal tunnel in seven (16%) patients (four bilateral). All the patients reported the occupational performance of repetitive hand movements; in addition, one patient had type 2 diabetes, and another patient reported history of traumatic injury of the affected arm. The recordings were similar at the two evaluations;○Peroneal nerves in seven (16%) patients; below the ankle in four patients, and at the level of the fibular head in three patients, including one patient with bilateral compromise. Two of them reported history of traumatic injury of the affected foot. At the second evaluation, the neuropathy resolved only in one patient with no comorbidities: a 33 year old female, who required 55 days of hospital stay (without mechanical ventilation), and had a 14.3% loss of her body mass at hospital discharge.Generalized myopathy was diagnosed in one patient, a 39 year old man, who required 28 days of hospital stay (without mechanical ventilation), receiving dexametasone and antibiotics (moxifloxacine, azitromicine and ceftriaxone).

#### 3.1.6. Barthel Index

At the two evaluations, all the participants reported independence to perform the basic activities of daily life, except for one patient (with generalized myopathy) who reported limitations for grooming.

#### 3.1.7. Six Minute Walk Test

At the beginning of the two evaluations, all the patients reported no dyspnea, but variable increase during the test (range 0 to 4). At the first evaluation, the distance walked by the patients was only 78.9% (95% C.I. 66.8%–91%) of the predicted distance, with oxygen saturation decrease to 89% (95% C.I. 80%–98%), tachypnea (mean rate ± standard deviation, 27 ± 3/min), and tachycardia (120 ± 22/min)) (Table 5). At the second evaluation, the absolute distance increased (paired “t” test, t = 5.81, *p* < 0.0001), the proportional distance from predicted increased (paired “t” test, t = 6.48, *p* < 0.0001), with less oxygen saturation decrease (paired “t” test, t = 3.30, *p* < 0.002). Moderate to strong linear correlations were observed on the walked distance, with age, muscle strength, body mass index at hospitalization and weight loss during hospitalization (Pearson’s r from 0.31 to 0.53, *p* < 0.05); additionally, moderate to strong correlations were observed between the proportional distance and the MRCs score (Pearson’s r = 0.46, *p* = 0.002), and the weight loss during hospitalization (Pearson’s r = 0.52, *p* < 0.0001).

#### 3.1.8. Berg Balance Scale

At the first evaluation, all participants reported no difficulty to stand or sit without support. However, they reported limitations on activities requiring leg strength: standing on one foot, standing with one foot in front of the other foot and reaching forward with outstretched arm (Table A2 of Appendix B). At the second evaluation, the scores improved (“t” test, t = 3.36, *p* = 0.001). Moderate linear correlations were observed between the total score at the first evaluation and the MRCs (Pearson´s r = −0.34, *p* = 0.025), and the BMI at hospital admission (Pearson´s r = 0.43, *p* = 0.03).

#### 3.1.9. Timed-Up-and-Go test

At the first evaluation, in all the patients, the mean time to perform the test was 8.69 ± 1.48 s. However, 12 patients performed the test in >10 s; they were 39 to 74 years old (six men/six women), all had neuromuscular abnormalities, three had sarcopenia and one had severe COVID-19. At the second evaluation, the time to perform the test decreased (8.05 ± 1.31 s) (paired “t” test, t = 3.29, *p* = 0.002); however, four patients performed the test in >10 s, and the three patients who had abandoned the study had performed the test in >10 s at the first evaluation. The time to perform the test at the first evaluation showed moderate linear correlation with age (Pearson´s r = 0.31, *p* = 0.04) and muscle strength of both the upper and lower limbs (Pearson´s r from 0.34 to 0.48, *p* < 0.03).

Analysis according to electrophysiology abnormalities showed that, compared with patients with no electrophysiology abnormalities, those with any abnormality were older (53.5 ± 9.2 versus 47.5 ± 8.37, t = 2.07, *p* = 0.04), had greater weight loss during hospital stay (10.8 ± 6.7 kg versus 4.6 ± 3.7 kg, t = 3.28, *p* = 0.002), and had greater weight gain after hospital discharge (9.4 ± 9.1 kg versus 2.8 ± 4.2 kg, t = 2.65, *p* = 0.01). At the two evaluations, the frequency of symptoms was similar in the two groups (*p* > 0.05). At the first evaluation, compared with patients with no electrophysiology abnormalities, patients with any abnormality had lower scores on the MRCs (57.1 ± 2.7 versus 58.8 ± 1.8, t = 2.13, *p* = 0.03) while, on the 6MWT, they walked a shorter distance (499.3 ± 77.0 m versus 589.9 ± 50.2 m, t = 4.09, *p* = 0.0001), and proportional distance (76.7% ± 10.41% versus 83.0% ± 6.9%, t = 2.11, *p* = 0.04); they also had a lower score on the Berg Balance Scale (52.2 ± 3.8 versus 55.2 ± 1.3, t = 2.98, *p* = 0.004). At the second evaluation, a difference persisted on the 6MWT, on the walked distance (545.6 ± 92.9 m versus 625.9 ± 53.8 m, t = 3.04, *p* = 0.004), while it was borderline on the Berg Balance Scale (“t” test, t = 1.98, *p* = 0.054).

### 3.2. Multivariate Analysis

The multivariate analysis of covariance on the scores of the Berg Balance Scale showed that it was related to the BMI at hospital admission (*p* = 0.01) and the walked distance during the 6MWT (*p* = 0.01) (beta values are described in Table 6). At the first evaluation, the whole model R was 0.69 (adjusted R^2^ = 0.40, F = 6.39, *p* = 0.0002); and at the second evaluation the whole model R was 0.65 (adjusted R^2^ = 0.33, F = 5.01, *p* = 0.001); with difference between the two evaluations (*p* = 0.047), which was related to the walked distance during the 6MWT (*p* = 0.009).

The multivariate analysis of covariance on the time to perform the Timed-Up-and-Go test showed that it was related to: the proportional weight loss at hospital discharge (*p* = 0.001); electrophysiology abnormalities (*p* = 0.02); sex (*p* = 0.004); the walked distance during the 6MWT (only at the first evaluation, *p* = 0.03, with a borderline overall result *p* = 0.053); and the handgrip strength, either of each hand or the normalized value, but particularly the absolute strength of the dominant hand (*p* = 0.007), all the patients were right handed (beta values are described in Table 6). At the first evaluation the whole model R was 0.77 (adjusted R^2^ = 0.52, F = 8.17, *p* < 0.0001); at the second evaluation, the whole model R was 0.62 (adjusted R^2^ = 0.27, F = 3.45, *p* = 0.0009); with difference between the two evaluations (*p* = 0.007), which was related to the handgrip strength of the right (dominant) hand (*p* = 0.004), and sex (*p* = 0.005) since men improved more than women.

## 4. Discussion

After moderate to severe COVID-19, patients can have neuromuscular abnormalities interfering with their mobility that require intended evaluation by standardized tools. The results suggest that high BMI and rapid weight loss during hospital stay are relevant factors for neuropathy. In this study, the 6MWT and handgrip dynamometry of the dominant hand were particularly useful to assess fitness, while the Timed-Up-and-Go test allowed quick evaluation of mobility.

Previous studies in patients with COVID-19 have suggested that virus-induced state of inflammation or overlapping comorbidities may contribute to nerve injuries [35]. In patients requiring intensive care, focal neurological deficits have been related to super-imposed mono-neuropathies of unknown etiology [36]; while in patients with diabetes, severe COVID-19 has been related to neuropathy symptoms and widespread sensory dysfunction [37]. In Italy, assessment of 102 convalescent patients who were admitted to a rehabilitation clinic showed electrophysiological abnormalities in 42.2% of patients, including peroneal mono-neuropathy in 8.8%, and multiple mono-neuropathy in 8.8% [38]. In this study, the majority of patients had moderate disease, and showed frequent peripheral neuropathy. The results showed that, in addition to a variety of individual predisposing factors mostly related to polyneuropathy and median nerve neuropathy, overall electrophysiological abnormalities were related to high BMI and rapid weight loss during hospitalization.

Clinical evidence supports a relationship between body weight and COVID-19 severity and its complications [39]. Additionally, weight loss is frequent in hospitalized patients with COVID-19 [40]; in a cohort study of 213 patients (73% hospitalized) weight loss > 5% was evident in 29% of patients, and it was related to systemic inflammation [40]. The findings of this study suggest that subclinical peripheral nerve impairment related to obesity and weight loss can become evident during the acute phase of moderate to severe COVID-19. The findings are consistent with previous studies showing that individuals with obesity are more likely to be diagnosed with carpal tunnel syndrome, related to fat or edema near the carpal tunnel compressing the median nerve [41]; while weight loss in a short period may modify the surroundings of the common peroneal nerve, provoking an entrapment in the peroneal tunnel [42]. Although metabolic effects of body weight and weight loss on the peripheral nervous system cannot be disregarded [43].

We also observed one patient with generalized myopathy after prolonged hospital stay, who received steroid therapy. In patients with severe COVID-19, direct effects of the infection on the muscle remain unclear. However, histopathology findings in skeletal muscle and peripheral nerves from 35 patients who died after COVID-19 has shown inflammatory/immune-mediated damage [44]. Additionally, myopathy and neuropathy have been related to critical illness and medical care; in 12 patients with severe COVID-19 and suspicion of myopathy or polyneuropathy, seven patients had signs of myopathy, and four patients had signs of sensory–motor axonal polyneuropathy, without distinctive features [45].

In this study, the symptoms reported by the patients were not always specifically related to the evidence of neuromuscular abnormalities. The most frequent symptom was fatigue, which is recognized as one of the main symptoms of post-acute COVID or Long COVID syndrome [46]. The persistence of fatigue is consistent with the post-infective fatigue syndrome that has been reported after a variety of infections [47]. Post-infectious fatigue syndrome is a subtype of chronic fatigue syndrome [48] that either follows an infection or is associated with a current infection, which cannot be explained by other medical or psychiatric conditions and has been present for at least six months (affecting daily functioning), and fulfill specific criteria for diagnosis [48]. Several theories about the etiology of fatigue after COVID-19 have been proposed, including cardiovascular, neuromuscular, neurological and psychosocial factors [47]; among them, cardiovascular factors have been associated with worsened adverse outcomes after COVID-19, including dysregulation of the autonomic nervous system [49].

The results of this study showed balance deficiencies mainly on activities requiring leg strength, but were also related to handgrip strength. We observed that both balance and leg strength recovered simultaneously through time, while in patients with neuromuscular abnormalities sub-optimal performance on the Timed-Up-and-Go test (>10 s) persisted. We observed that the results on the two mobility tests (Berg Balance Scale and Timed-Up-and-Go test) were related to both the 6MWT and the muscle strength. However, the recovery of quadriceps strength was related to improvement on the Berg Balance Scale, while the handgrip strength was mainly related to the time to perform the Timed-Up-and-Go test. A previous study has shown that handgrip strength is related to hospitalization due to COVID-19, independent of age, sex, and health-related conditions [50], while it is also associated with cardiovascular and all-cause mortality events [51].

Walking and postural transitions are among the basic motor tasks that every person must perform several times a day. In this study, we observed that patients discharged from hospital, with no history or evidence of neurological deficits, may have sub-clinical neuropathy and muscle weakness interfering with balance and mobility. These findings suggest that, in order to select adequate rehabilitation strategies, individual needs should be intentionally evaluated, including neuromuscular and balance tests.

The main limitation of this study was the small, convenient sample. The sample size was estimated to assess relationships between variables, while reducing the variability that could be introduced by some recognized co-factors (i.e., limited respiratory capacity and neurological disorders). Another limitation was that we mainly evaluated patients with moderate disease, only four had severe disease, and none had mild disease; the characteristics of each subgroup may imply variability on both neuromuscular abnormalities and mobility. Although the selection bias may have driven to overestimation of performance, it allowed us to identify associations between neuromuscular abnormalities, mobility and the influence of several cofactors; particularly the association among high BMI at hospital admission and weight loss during hospital stay with neuromuscular abnormalities and functional performance, which has not been reported previously in patients with COVID-19. Further studies with broader selection criteria may show the full spectrum of neuromuscular abnormalities and its consequences on mobility.

## 5. Conclusions

In adults, after hospital discharge due to moderate to severe COVID-19, neuromuscular abnormalities may contribute to mobility decrease; while both BMI at admission and weight loss during hospital stay may contribute to neuromuscular abnormalities. These abnormalities may be identified only after intended evaluations, including electromyoneurography and mobility tests. In addition, apart from the 6MWT, handgrip dynamometry and the Timed-Up-and-Go test could be useful tools for the assessment and follow-up of patients with COVID-19.

## Figures and Tables

**Figure 1 medicina-58-01401-f001:**
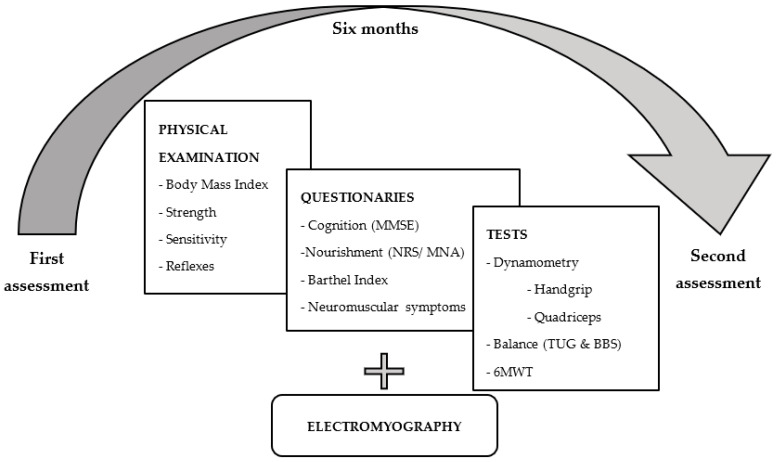
Assessments were performed two times, with six months in between, to 43 patients the first time, and 40 patients the second time. MMSE: Mini-Mental State Examination; NRS: Nutritional Risk Screening; MNA: Mini Nutritional Assessment; TUG: Timed-Up-and-Go Test; BBS: Berg Balance Scale; 6MWT: Six Minute Walk Test.

**Figure 3 medicina-58-01401-f003:**
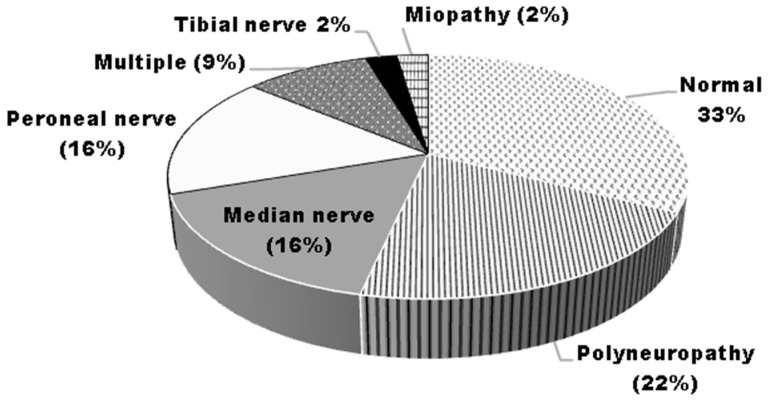
Frequency of electrophysiology abnormalities in the 43 patients participating in the study.

**Table 1 medicina-58-01401-t001:** General characteristics of the 43 patients included in the study.

Characteristics	Women (n = 18)		Men (n = 25)		All (n = 43)	
	Mean (S.D.)	Range	Mean (S.D.)	Range	Mean (S.D.)	Range
Years of age	51 (9)	32–60	51 (9)	38–74	51 (9)	32–74
Days from onset to hospitalization	9 (4)	5–20	11 (5)	2–28	10 (5)	2–28
Days in hospital	16 (11)	7–55	16 (12)	7–61	16 (11)	7–61
	*n*	%	*n*	%	*n*	%
Type 2 diabetes	5	27	4	16	9	20
Systemic high blood pressure	6	33	7	28	13	30
B.M.I. ≥ 25 at hospital admission	15	83	21	21	36	83
B.M.I. ≥ 25 at hospital discharge	10	55	12	12	22	51

S.D.: standard deviation; B.M.I.: body mass index.

**Table 2 medicina-58-01401-t002:** Frequency of the symptoms reported by 43 patients at the first evaluation, and 40 patients at the second evaluation.

	Evaluation I			Evaluation II		
Symptoms	Women (n = 18)	Men (n = 25)	All (n = 43)	Women (n = 17)	Men (n = 23)	All (n = 40)
Myalgia	11 (61%)	8 (32%)	19 (44%)	7 (41%)	6 (26%)	13 (32%)
Fatigue	15 (83%)	18 (72%)	33 (77%)	11 (65%)	10 (43%)	21 (52%)
Muscle spasms/twitches/tremors	7 (39%)	5 (20%)	12 (28%)	5 (29%)	2 (9%)	7 (17%)
Numbness/tingling/burning sensations	8 (44%)	6 (24%)	14 (33%)	5 (29%)	2 (9%)	7 (17%)

**Table 3 medicina-58-01401-t003:** Mean and standard deviation of the mean of the muscle strength on the handgrip test and the quadriceps dynamometry of 43 patients at the first evaluation, and 40 patients at the second evaluation.

Test	Evaluation I			Evaluation II		
	Right	Left	Average	Right	Left	Average
**Men**						
Handgrip (kg)	33.7 ± 7.0	32.0 ± 7.7	32.9 ± 7.4	32.5 ± 6.3	30.9 ± 5.7	31.7 ± 6.1
Normalized handgrip (kg)	-	-	0.42 ± 0.10	-	-	0.38 ± 0.09
Quadriceps (kg)	34.6 ± 10.1	33.8 ± 10.4	34.2 ± 10.3	40.5 ± 7.4	39.9 ± 6.8	40.2 ± 7.1
Normalized quadriceps	-	-	0.43 ± 0.12	-	-	0.49 ± 0.11
**Women**						
Handgrip (kg)	21.7 ± 5.1	21.7 ± 3.9	21.7 ± 4.5	22.7 ± 6.2	22.9 ± 5.3	22.8 ± 5.7
Normalized handgrip (kg)	-	-	0.31 ± 0.08	-	-	0.31 ± 0.10
Quadriceps (kg)	26.6 ± 8.4	24.8 ± 7.9	25.7 ± 8.2	29.9 ± 6.9	29.5 ± 7.5	29.7 ± 7.2
Normalized quadriceps	-	-	0.36 ± 0.10	-	-	0.40 ± 0.10

**Table 4 medicina-58-01401-t004:** Mean and standard deviation of the mean of the latency and amplitude of neuro-conduction tests, and the latency of F wave and H reflex of 43 patients at the first evaluation and 40 patients at the second evaluation.

		Evaluation I				Evaluation II			
		Latency (milliseconds)		Amplitude (microamperes)		Latency (milliseconds)		Amplitude (microamperes)	
	Nerve	Right	Left	Right	Left	Right	Left	Right	Left
	**Sensory**								
Men	Cubital	3.2 ± 0.2	3.2 ± 0.2	22.2 ± 4.6	21.5 ± 5.9	3.2 ± 0.1	3.2 ± 0.1	25.4 ± 2.1	25.5 ± 2.5
	Median	3.4 ± 0.5	3.4 ± 0.5	24.6 ± 11.5	26.0 ± 11.7	3.5 ± 0.5	3.4 ± 0.5	27.3 ± 11.3	28.2 ± 10.5
	Radial	3.0 ± 0.2	3.0 ± 0.2	9.5 ± 2.5	9.9 ± 2.5	3.0 ± 0.1	3.0 ± 0.1	10.8 ± 2.1	10.7 ± 1.9
	Sural	2.6 ± 0.4	2.8 ± 0.3	18.6 ± 5.4	17.0 ± 6.3	2.9 ± 0.3	2.9 ± 0.3	17.2 ± 4.8	17.7 ± 5.0
Women	Cubital	3.0 ± 0.2	2.9 ± 0.3	27.7 ± 12.4	30.2 ± 15.4	3.0 ± 0.2	3.0 ± 0.2	30.1 ± 11.7	31.3 ± 13.5
	Median	3.5 ± 0.7	3.4 ± 0.7	27.4 ± 17.7	27.4 ± 15.2	3.6 ± 0.7	3.6 ± 0.5	28 ± 16.1	30.1 ± 15.2
	Radial	2.9 ± 0.3	2.9 ± 0.3	12.1 ± 5.1	12.0 ± 4.0	2.9 ± 0.2	2.9 ± 0.2	12.1 ± 4.0	12.2 ± 3.6
	Sural	2.7 ± 0.3	2.8 ± 0.3	18.9 ± 9.7	17.1± 7.5	2.9 ± 0.3	2.9± 0.3	18.8 ± 8.7	18.9 ± 9.8
		**Amplitude** **(milliamperes)**		**Velocity** **(m/s)**		**Amplitude** **(milliamperes)**		**Velocity** **(m/s)**	
		**Right**	**Left**	**Right**	**Left**	**Right**	**Left**	**Right**	**Left**
	**Motor**								
Men	Cubital	4.3 ± 1.5	4.8 ± 1.7	59.5 ± 5.3	58.5 ± 4.5	5.0 ± 1.2	5.3 ± 1.3	59.1 ± 4.5	58.6 ± 3.4
	Median	5.7 ± 1.7	5.9 ± 1.5	54.4 ± 2.4	54.6 ± 2.7	7.0 ± 1.3	7.0 ± 1.2	55.4 ± 2.2	56.0 ± 2.3
	Peroneal	2.7 ± 1.3	2.6 ± 1.5	50.2 ± 5.8	48.8 ± 5.1	3.2 ± 1.1	3.1 ± 1.3	49.1 ± 4.2	48.5 ± 4.8
	Tibial	4.6 ± 2.3	4.3 ± 2.0	49.0 ± 5.2	48.9 ± 5.5	5.0 ± 2.1	4.9 ± 2.0	48.8 ± 4.0	48.4 ± 4.4
Women	Cubital	4.1 ± 1.6	4.2 ± 1.6	60.0 ± 5.8	59.8 ± 6.8	4.7 ± 1.4	4.9 ± 1.5	59.6 ± 5.2	59.8 ± 5.1
	Median	5.1 ± 2.3	5.7 ± 1.7	53.8 ± 5.5	55.4 ± 5.5	6.2 ± 2.3	7.0 ± 0.8	54.2 ± 5.5	55.7 ± 5.6
	Peroneal	2.6 ± 0.9	2.8 ± 1.5	51.2 ± 4.6	50.3 ± 6.9	3.3 ± 0.9	3.3 ± 1.4	51.1 ± 3.5	49.9 ± 6.0
	Tibial	4.4 ± 1.5	4.3 ± 1.6	49.7 ± 8.7	49.2 ± 8.3	4.7 ± 1.6	4.7 ± 1.5	49.8 ± 7.6	49.5 ± 7.1
		**F-Wave** **Latency** **(milliseconds)**		**H-Reflex** **Latency** **(milliseconds)**		**F-Wave** **Latency** **(milliseconds)**		**H-Reflex** **Latency** **(milliseconds)**	
		**Right**	**Left**	**Right**	**Left**	**Right**	**Left**	**Right**	**Left**
Men									
	Cubital	27.1 ± 1.9	27.2 ± 1.7	-	-	26.2 ± 5.3	27.3 ± 1.7	-	-
	Median	27.1 ± 1.6	27.1 ± 1.7	16.9 ± 2.0	17.1 ± 2.0	27.2 ± 1.6	27.3 ± 1.7	16.8 ± 0.6	16.8 ± 0.6
	Tibial	48.4 ± 3.7	48.4 ± 3.8	30.7 ± 2.9	30.2 ± 2.5	48.4 ± 3.7	48.4 ± 3.7	30.0 ± 1.5	29.9 ± 1.5
Women									
	Cubital	25.7 ± 2.6	25.6 ± 2.6	-	-	25.8 ± 2.6	25.7 ± 2.6	-	-
	Median	25.9 ± 2.5	25.5 ± 2.2	16.3 ± 2.0	16.3 ± 2.3	26.1 ± 2.4	25.8 ± 2.3	16.7 ± 1.6	16.5 ± 2.1
	Tibial	44.5 ± 2.5	45.3 ± 3.5	30.0 ± 3.2	29.7 ± 2.2	44.8 ± 2.5	44.8 ± 2.6	30.1 ± 3.2	29.7 ± 2.2

m/s: meters per second.

**Table 5 medicina-58-01401-t005:** Mean and standard deviation of the mean of the cardiorespiratory measurements during the 6 Minute Walk Test in 43 patients at the first evaluation and 40 patients at the second evaluation.

Variable	Evaluation I	Evaluation II
Distance (meters)	531 ± 80.1	576 ± 87.8
Proportional distance from predicted (%)	78.9 ± 9.6	86 ± 11.2
Initial oxygen saturation (%)	94 ± 2	95 ± 2
Minimum oxygen saturation (%)	89 ± 4	92 ± 2
Initial respiratory rate (breaths per minute)	20 ± 2	19 ± 1
Respiratory rate at the end (breaths per minute)	27 ± 3	25 ± 2
Initial heart rate (beats per minute)	80 ± 16	77 ± 12
Maximum heart rate (beats per minute)	120 ± 22	125 ± 12
Initial arterial blood pressure (mmHg)	123 ± 13/79 ± 11	126 ± 15/79 ± 11
Arterial blood pressure at the end (mmHg)	134 ± 19/82 ± 10	134 ± 17/81 ± 8

**Table 6 medicina-58-01401-t006:** Beta values and 95% C.I. of the beta values of the variables included in the general linear model on the time to perform the Timed-Up-and-Go test and the score on the Berg Balance Scale.

Test	Factors	Evaluation I		Evaluation II	
		Beta (ß)	95% C.I.	Beta (ß)	95% C.I.
Berg Balance Scale	MRC scale score	0.10	−0.23–0.42	−0.02	−0.36–0.33
	Handgrip (dominant hand)	−0.17	−0.62–0.29	−0.36	−0.84–0.13
	6MWT distance	0.52	0.17–0.88	0.31	−0.07–0.69
	Body mass index at hospital admission	−0.28	−0.58–0.02	−0.48	−0.80–−0.17
	Sex	−0.02	−0.41–0.37	−0.32	−0.73–0.10
Timed-Up-and-Go test	Handgrip (dominant hand)	−0.76	−1.14–−0.39	−0.30	−0.77–0.16
	Distance on the 6MWT	−0.36	−0.68–−0.04	−0.29	−0.68–0.11
	Proportional weight loss	−0.48	−0.76–−0.19	−0.51	−0.86–−0.16
	Sex	−0.73	−1.08–−0.39	−0.33	−0.76–0.10
	Electrophysiology abnormalities	−0.24	−0.52–0.04	−0.40	−0.75–−0.06
	Sex*electrophysiology abnormalities	−0.12	−0.36–0.12	−0.20	−0.50–0.10

## Data Availability

The data are contained within the article and Appendix B. The datasets are available from the corresponding author on reasonable request.

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
