# Peer review of "Body Weight May Have a Role on Neuropathy and Mobility after Moderate to Severe COVID-19: An Exploratory Study"

_medicina, 2022, doi:10.3390/medicina58101401_

Round 1
Reviewer 1 Report
Dear authors,
The study is very interesting, but I propose improvements.
Introduction
It is very short. I suggest to introduce the pathology, clinical manifestations, health care on admission, physical, pulmonary and cardiac sequelae'. I suggest introducing the pathology, clinical manifestations, health care on admission, physical, pulmonary and cardiac sequelae.
Table 1
It is not very interpretable.
Mixes ranges with averages.
I suggest including in separate columns: n, means, standard deviations.
Outcomes measures
I suggest including a section on outcome measures, briefly explaining each measure and its relationship to the object of study.
The discussion should be the longest part of an article.
It is necessary to compare and discuss the findings of this study with previous research in patients with COVID or other similar pathologies.
Kind regards,

Author Response
The study is very interesting, but I propose improvements.
WE THANK THE REVIEWER FOR THE PURPOUSEFUL COMMENTS.
Introduction
It is very short. I suggest to introduce the pathology, clinical manifestations, health care on admission, physical, pulmonary and cardiac sequelae'.
THE INTRODUCTION HAS BEEN EDITED TO INCLUDE THE MAIN COMPLICATIONS OF THE DISEASE, SUCH AS CARDIOVASCULAR COMPLICATIONS
Table 1 It is not very interpretable. Mixes ranges with averages. I suggest including in separate columns: n, means, standard deviations.
TABLE 1 HAS BEEN EDITED ACCORDINGLY
Outcomes measures. I suggest including a section on outcome measures, briefly explaining each measure and its relationship to the object of study.
WE HAVE EDITED THE DESCRIPTION OF THE PROCEDURES TO EXPLAIN EACH MEASURMENT.
WE HAVE INCLUDED A FIGURE TO SIMPLIFY THE DESCRIPTION OF THE METHODS.
The discussion should be the longest part of an article. It is necessary to compare and discuss the findings of this study with previous research in patients with COVID or other similar pathologies.
WE THANK THE REVIEWER FOR THE RECOMMENDATION. THE DISCUSSION HAS BEEN EDITED AND EXPANDED TO INCLUDE MORE STUDIES. HOWEVER, SINCE THE STUDY WAS PERFORMED IN A SELECTED SAMPLE (NO DYSPNEA AND NO NEUROMUSCULAR/ NEUROLOGICAL/ORTHOPEDIC DISORDERS) AND THERE ARE NO STUDIES EVALUATING THE SAME RELATIONSHIPS, THE AUTHORS PREFER THE DISCUSSION TO BE BRIEF AND CONCISE.
Reviewer 2 Report
1. Its existing title's capitalization should be updated to follow the MDPI format.
2. Title is recommended to change not using commas “,:”, but colon “:”
3. In the abstract section, quantitative data must be included.
4. As your abstract's final sentence, include a "take-home" message.
5. Keywords should be rearranged alphabetically.
6. In the present form, nothing really novel. The current study appears to be a replication or modified study according to the lack of novelty. The authors must extensively describe the novel their work is. This work should be rejected due to a serious concern.
7. The work, novelty, and limitations of similar prior studies must be explained in the introduction section to highlight the research gaps that the current study aims to fill.
8. Explain specifically the objective of the present study in the last paragraph of the introduction section.
9. Body wight as indicatorid by body mass index have been studied that corelate with other health issue for implant user make the risk of implant failure become higher along with weigher patient as reported by Ammarullah et al. is a vital topic that authors must provide in the introduction and/or discussion section. Additionally, the suggested reverence should be taken to substantiate this explanation as follows: Ammarullah, M. I.; Santoso, G.; Sugiharto, S.; Supriyono, T.; Kurdi, O.; Tauviqirrahman, M.; Winarni, T. I.; Jamari, J. Tresca Stress Study of CoCrMo-on-CoCrMo Bearings Based on Body Mass Index Using 2D Computational Model. Jurnal Tribologi 2022, 33, 31–8. https://jurnaltribologi.mytribos.org/v33/JT-33-31-38.pdf
10. To help the reader grasp the study's workflow more easily, the authors could include more visuals to the materials and methods section in the form of figures rather than sticking with the text that now predominates.
11. What is the baseline of participant selection? Is there any protocol, standard, or basis that has been followed? It is unclear since the patient is very heterogeneous with a small number. The resonance involved impacts the present result makes this study flaws. One major reason for rejecting this paper.
12. An assessment of the findings with similar past investigations is recommended.
13. Another limitation apart from that have been mentioned in line 324-329?
14. In the conclusion, please explain the further research.
15. The reference should be enriched with literature from the last five years. Literature published by MDPI is strongly recommended.
16. In the entire manuscript, the authors occasionally constructed paragraphs with just one or two phrases, which made the explanation difficult to understand. To make their explanation a full paragraph, the authors should expand it. It is advised to use at least three sentences in a paragraph, with the primary sentence coming first and the supporting sentences coming after.
17. Due to grammatical and language issues, the authors need to proofread the present work. This problem would use MDPI English editing service.
18. Please review and confirm that the writers followed the MDPI format exactly, edit the current form, and recheck in addition to the other issues that have been mentioned.
Author Response
- Its existing title's capitalization should be updated to follow the MDPI format.
THANK YOU FOR THE COMMENT, THE TITLE HAS BEEN CAPITALIZED.
- Title is recommended to change not using commas “,:”, but colon “:”
THANK YOU FOR THE COMMENT, THE COMA HAS BEEN REMOVED.
- In the abstract section, quantitative data must be included &
- As your abstract's final sentence, include a "take-home" message.
THE ABSTRACT HAS BEEN EDITED ACCORDINGLY.
- Keywords should be rearranged alphabetically.
THE KEY WORDS HAVE BEEN REARRENGED.
- In the present form, nothing really novel. The current study appears to be a replication or modified study according to the lack of novelty. The authors must extensively describe the novel their work is. This work should be rejected due to a serious concern.
- The work, novelty, and limitations of similar prior studies must be explained in the introduction section to highlight the research gaps that the current study aims to fill.
THE INTRODUCTION HAS BEEN EDITED. THERE MAY BE A CONFUSIÓN, SINCE THE REVIEW OF THE LITERATURE SHOWED NO PREVIOUS STUDY WITH SIMILAR PROCEDURES AND OUTCOMES. EVEN MORE, THERE IS NO PREVIOUS STUDY SHOWING THE INTERACTION BETWEEN BMI/WEIGHT LOSS AND ELECTROPHYSIOLOGICAL ABNORMALITIES ON MOBILITY TESTS, AFTER HOSPITALIZATION DUE TO COVID-19.
- Explain specifically the objective of the present study in the last paragraph of the introduction section.
IN THE LAST PARAGRAPH OF THE INTRODUCTION, THE OBJETIVE TO DESIGN THE STUDY HAS BEEN EDITED TO EMPHAZISE THE PRIMARY AND SECONDARY AIMS OF THE STUDY
- Body wight as indicatorid by body mass index have been studied that corelate with other health issue for implant user make the risk of implant failure become higher along with weigher patient as reported by Ammarullah et al. is a vital topic that authors must provide in the introduction and/or discussion section. Additionally, the suggested reverence should be taken to substantiate this explanation as follows: Ammarullah, M. I.; Santoso, G.; Sugiharto, S.; Supriyono, T.; Kurdi, O.; Tauviqirrahman, M.; Winarni, T. I.; Jamari, J. Tresca Stress Study of CoCrMo-on-CoCrMo Bearings Based on Body Mass Index Using 2D Computational Model. Jurnal Tribologi 2022, 33, 31–8. https://jurnaltribologi.mytribos.org/v33/JT-33-31-38.pdf
THERE MAY BE A MISSUNDERSTANDING, SINCE THE REFERENCE PROVIDED CORRESPONDS TO A STUDY AIMED TO INVESTIGATE TRESCA STRESS IN METAL-ON-METAL HIP IMPLANT. THIS STUDY DID NOT INCLUDE ANY PARTICIPANT WITH PREVIOUS NEUROMUSCULAR DISORDERS, EVEN LESS WITH THE NEED OF A HIP IMPLANT.
- To help the reader grasp the study's workflow more easily, the authors could include more visuals to the materials and methods section in the form of figures rather than sticking with the text that now predominates.
WE THANK THE REVIEWER FOR THE COMMENT AND WE HAVE INTRODUCED TWO ADDITIONAL FIGURES .
- What is the baseline of participant selection? Is there any protocol, standard, or basis that has been followed? It is unclear since the patient is very heterogeneous with a small number. The resonance involved impacts the present result makes this study flaws. One major reason for rejecting this paper.
THE PARAGRAPH DESCRIBING THE PARTICIPANTS HAS BEEN EDITED TO CLARIFY THIS POINT.
- An assessment of the findings with similar past investigations is recommended.
WE HAVE REVISED THE LITERATURE AND UPDATED THE DISCUSSION BUT NO SIMILAR STUDIES WERE FOUND
- Another limitation apart from that have been mentioned in line 324-329?
THE LIMITATION PARAGRAPH HAS BEEN EDITED AND EXPANDED ACCORDINGLY.
- In the conclusion, please explain the further research.
FURTHER REASEARCH HAS BEEN PROPOSED JUST BEFORE THE CONCLUSIONS
- The reference should be enriched with literature from the last five years. Literature published by MDPI is strongly recommended.
WE HAVE INCLUDED ADDITIONAL REFERENCES, AS REQUIERED BY THE ENRICHMENT OF THE TEXT, WHICH WERE SELECTED BY THEIR CONTENT.
- In the entire manuscript, the authors occasionally constructed paragraphs with just one or two phrases, which made the explanation difficult to understand. To make their explanation a full paragraph, the authors should expand it. It is advised to use at least three sentences in a paragraph, with the primary sentence coming first and the supporting sentences coming after.
THE MANUSCRIPT WAS WRITTEN ACCORDING TO THE SCIENTIFIC CONTENT OF EACH SECTION. IT HAS BEEN REVISED TO ENSURE THAT WHENEVER A LIST IS DESCRIBED, IT IS BULLETED TO DISTINGUISH IT FROM A PARAGRAPH.
- Due to grammatical and language issues, the authors need to proofread the present work. This problem would use MDPI English editing service.
THE MANUSCRIPT HAS BEEN REVISED.
- Please review and confirm that the writers followed the MDPI format exactly, edit the current form, and recheck in addition to the other issues that have been mentioned.
THE MDPI PAPER FORMAT (WORD) WAS USED TO CONSTRUCT THE MANUSCRIPT.
Round 2
Reviewer 1 Report
Dear Authors,
The revised article incorporates major improvements.
Kind regards,
Author Response
We thank you for your recommendations.
Best regards
Reviewer 2 Report
Reviewers greatly appreciate the efforts that have been made by the author to improve the quality of their articles after peer review. I reread the author's manuscript and further reviewed the changes made along with the responses from previous reviewers' comments. Unfortunately, the authors failed to make some of the substantial improvements they should have made making this article not of decent quality with biased, not cutting-edge updates on the research topic outlined. In addition, the author also failed to address the previous reviewer's comments, especially on comments number 6, 7, and 9. With all due respect, the reviewer opposed this article to be published and must be rejected. Thank you very much for the opportunity to read the author's current work.
Author Response
We thank the reviewr for the comments. However, the reference on hip-implants is not appropriate for this study; other references were included according to their content, including two studies that were published in mdpi journals . The recent revision of the international literature did not show any study with design and outcomes similar to our study,